# Tangram-Splatting: Optimizing 3D Gaussian Splatting Through Tangram-inspired Shape Priors

## ABSTRACT

As the growth of VR and AR industry, 3D reconstruction has become a more and more important topic in multimedia. Although 3D Gaussian Splatting is the state-of-the-art method of 3D Reconstruction, it needs a large number of Gaussians to fit a 3D scene due to the Gibbs Phenomenon. The pursuit of compressing 3D Gaussian Splatting and reducing memory overhead has long been a focal point. Embarking on this trajectory, our study delves into this domain, aiming to mitigate these challenges. Inspired by tangram, a Chinese ancient puzzle, we introduce a novel methodology (Tangram-Splatting) that leverages shape priors to optimize 3D scene fitting. Central to our approach is a pioneering technique that diversifies Gaussian function types while preserving algorithmic efficiency. Through exhaustive experimentation, we demonstrate that our method achieves a remarkable average reduction of 62.4% in memory consumption used to store optimized parameters and decreases the training time by at least 10 minutes, with only marginal sacrifices in PSNR performance, typically under 0.3 dB, and our algorithm is even better on some datasets. This reduction in memory burden is of paramount significance for real-world applications, mitigating the substantial memory footprint and transmission burden traditionally associated with such algorithms. Our algorithm underscores the profound potential of Tangram-Splatting in advancing multimedia applications.

## CCS CONCEPTS

• **Computing methodologies** → **Reconstruction**.

## KEYWORDS

Multimedia, 3D Gaussian Splatting, Signals Processing

## 1 INTRODUCTION

In recent years, 3D reconstruction and novel view synthesis tasks have gained increasing popularity in multimedia, driven by the growing demand for Virtual Reality (VR) and Augmented Reality (AR) applications [13, 31] which is an important topic in multimedia [1, 15, 20, 22, 36]. Traditional methods for such tasks, *e.g.*, Structure from Motion (SfM) [26] and Multi-View Stereo (MVS) [12, 32, 33, 35], have yielded impressive results. Additionally, recent advancements such as Neural Radiance Field (NeRF) [21] and 3D Gaussian Splatting (3DGS) [14] have further pushed the boundaries

*ACM MM, 2024, Melbourne, Australia*
© 2024 Copyright held by the owner/author(s). Publication rights licensed to ACM.
ACM ISBN 978-x-xxxx-xxxx-x/YY/MM
https://doi.org/10.1145/nnnnnnn.nnnnnnn

of performance in these tasks. However, despite its advancements, 3DGS [14] still exhibits inherent limitations. The Gibbs phenomenon [7] inherent in the Gaussian Transform implies that accurately representing a scene requires an infinite number of Gaussians, which is impractical to be realized. Consequently, Gaussian-based representations have an upper bound of its performances.

In this study, we introduce Tangram-Splatting, a novel approach aimed at providing a more precise and efficient representation for 3D scenes. This method exhibits promising potential for enhancing 3D reconstruction tasks and may serve as inspiration for future research in the field. Inspired by previous works [4, 9, 17, 29], we recognize the diverse range of Gaussian-based functions, such as Gaussian Mixture functions, Difference of Gaussian functions, and Generalized Exponential Function, each offering unique advantages. For instance, Generalized Exponential Functions are better to converge on rectangular waves, Difference of Gaussian functions are better to converge on triangular waves and others are better to converge on a different shape. Tangram, an ancient Chinese puzzle which aims at filling a specific kind of shape by using different shapes of geometry, inspires us to leverage the advantages of those functions and combine them together to better fit the 3D scene. According to [10], directly changing the orginal Gaussian representation to a new Gaussian-based function is non-trivial, especially for the rasterization process. In order to reserve the original rasterization process, we use maximum likelihood estimation to represent different functions with an adaptive multiplication matrix, instead of directly changing the original Gaussian rasterization process. Different parts of a 3D scene require different kinds of shape priors, which means that we set strong condition to apply certain kind of Gaussian function to reconstruct a certain space based on the prior information about the eigenvalues of vanilla Gaussian. In order to search for the best fit of the 3D scene, we change the clone procedures of 3DGS [14] and set a brand new criterion to allocate the types of Gaussians in 3DGS [14].

Extensive experiments show that our approach improves 3DGS [14] approximately 62.4% in terms of memory efficiency while maintaining comparable image quality and training speed. In fact, our method performs even faster than 3DGS [14] for at least 10 minutes in several cases. Furthermore, we conducted experiments to validate the proposed theories.

The main contributions are summarized as follows:

(1) To the best of our knowledge, we are the first to leverage various types of Gaussians for 3D scene reconstruction with promising performance.

(2) By harnessing a new cloning technique, we successfully allocate the types of Gaussians while not sacrificing the speed of running the procedures which has saved almost 62.4% of the points that 3DGS [14] originally needed and decreasing at least 10 minutes of training time.

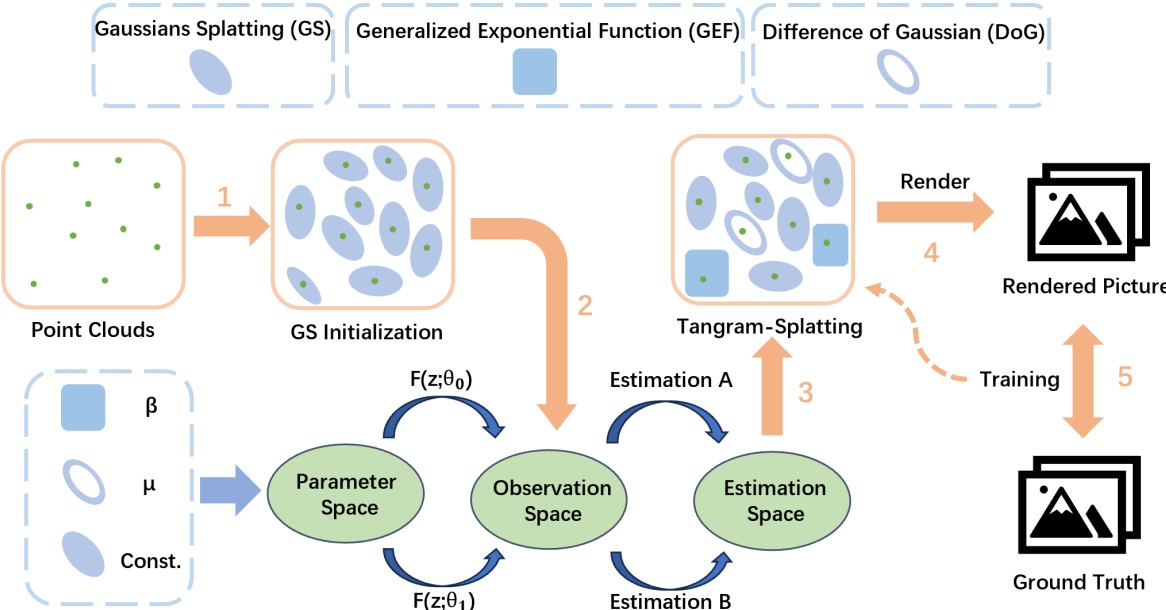

**Figure 1: Illustration of the proposed method. We have shown the big picture of our method in the figure. We argue that only using one particular type of Gaussian function is not a compact representation. As a result, we leverage 3 types of Gaussian-based functions, including vanilla Gaussian function (GS), Generalized Exponential function (GEF), Difference of Gaussian (DoG) to reconstruct a 3D scene. However, directly using 2 new functions as basis function is non-trivial, particularly we may face a drawback in the speed of rasterization process. As a result, we use the maximum likelihood estimation method to diversify the Gaussian basis function while maintaining the original rasterization methods. Our method can reduce the memory cost for about 62.4% compared to 3DGS [14], decrease the training time for about 10 minutes while maintaining the equivalent PSNR performance.**

(3) We have introduced a signal estimation framework aimed at reconstructing a 3D scene using various types of base functions, significantly enhancing the practical application of our work.

## 2 RELATED WORK

### 2.1 3D Gaussian Splatting

3D reconstruction and rendering has gained increasing attention. Algorithms include Neural Radiance Field [21, 34, 37], Structure from Motion [28] and other works [14, 30] have tremendously improve the performance of 3D reconstruction. Among them, 3DGS [14] is the state-of-the-art algorithm in 3D rendering and reconstruction field which has gained great popularity recently. It represents a 3D scene by lots of Gaussians and leverages rasterization methods and an adaptive control method to aprroximate a 3D scene. However, a lot of works [5, 18, 19, 24, 25] have claimed its bottleneck in memory cost and claim that it is not a compact representation. However, those works mainly focus on how to prune the trivial Gaussian and how to compress the parameters. In this work, we are going to refine the representation part. We claim that Gaussian function itself is not compact enough due to the Gibbs Phenomenon and we then design a brand new mechanism which can better represent a realistic 3D scene with less points needed.

### 2.2 Gaussian Transform

Inspired by lots of works [3, 8, 27], especially Generalized Exponential Splatting (GES) [10], we are interested in leveraging signals and systems perspectives to analyze the properties of 3DGS [14]. Gaussian Transform is a long studied signal processing field especially in the last century, which aimed at analyzing the properties when trying to use Gaussian functions to approximate a signal. We have found out that the last century's Gaussian Transform is so useful for analyzing 3DGS [14] and that, there is nothing new under the sun.

## 3 THEORETICAL ANALYSES

In this section, we present theoretical analyses of our framework. As we present in Fig. 1, we try to leverage different types of Gaussians to approximate the 3D scene in order to have a more compact representaion. The theoretical analyses involved in our approach will be described in detail below.

### 3.1 Properties of Different types of Basis Functions

To better understand the rationale behind employing diverse basis functions as opposed to relying solely on a single Gaussian function, as in the 3DGS algorithm [14], or a singular Generalized Exponential Function, as utilized in the GES algorithm [10], it is imperative

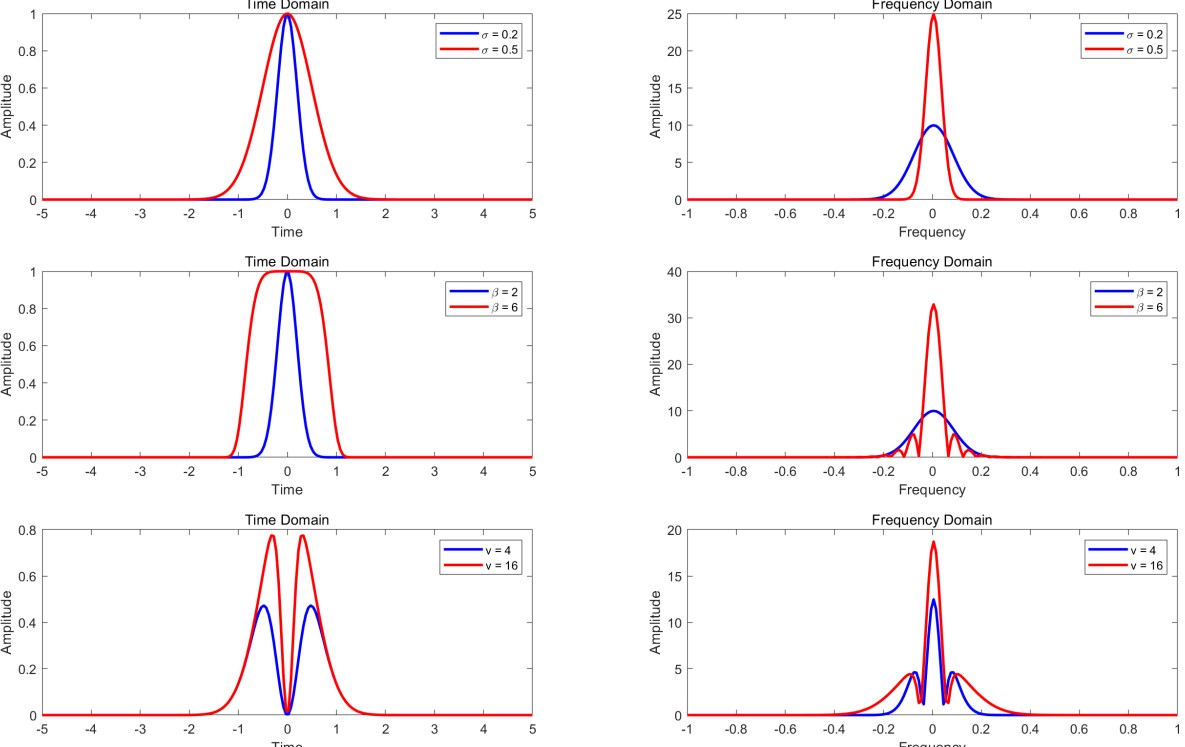

**Figure 2: The time domain and frequency domain of the three functions.**

to delve into the distinctive properties inherent in various basis functions. This strategic approach enables the reconstruction of scenes in accordance with localized conditions. To streamline our analysis, we embark on an exploration of the properties exhibited by various basis functions in a one-dimensional context.

**Gaussian Function.** The spatial domain expression and Fourier transform of a Gaussian function with a mean of $\mu$ and a variance of $\sigma^2$ is shown below:

$$\frac{1}{\sqrt{2\pi}\sigma}e^{-\frac{(x-\mu)^2}{2\sigma^2}} \leftrightarrow e^{-i\omega\mu}e^{-\frac{\omega^2\sigma^2}{2}}. \tag{1}$$

As depicted in Fig. 2, the Fourier transform of a Gaussian signal preserves its Gaussian nature. Notably, the temporal energy distribution of a Gaussian signal is predominantly concentrated within a range offset by $3\sigma$ from its mean $\mu$. Conversely, in the frequency domain, the energy of the signal is concentrated within a range offset by $\frac{3}{\sigma}$, centered around 0. Due to its inherent low-pass filtering characteristics, the Gaussian function necessitates an increase in the number of Gaussian components and the adoption of smaller Gaussian spheres by 3DGS [14] to effectively capture scene details. However, this approach inevitably leads to redundancy, inefficient storage utilization, and diminished computational efficiency.

**Generalized Exponential Function (GEF).** The one-dimensional expression of the Generalized Exponential Function (GEF), as described in [10], is $\frac{1}{\sqrt{2\pi}\sigma}e^{-\frac{(x-\mu)^\beta}{2\sigma^2}}$.

However, it is worth noting that the Fourier transform of GEF might lack an analytical solution, contingent upon the parameter

$\beta$ within the exponential term. This feature may contribute to the instability observed during the reconstruction process, as noted in [10]. For simulation purposes, specific parameter values were selected, and the outcomes are delineated in Fig. 2.

The one-dimensional simulation results from [10] indicate that GEF exhibits favorable performance when applied to smooth signals (*e.g.*, Parabolic Function, Gaussian Function), requiring fewer instances ($N$) to minimize fitting errors. However, when being confronted with sharp signals (*e.g.*, Triangle Function, Exponential Function), GEF tends to exhibit non-convergence. This observation implies that while GEF is adept at fitting smoother regions in 3D reconstruction, it may not yield optimal results for abrupt regions. Consequently, the introduction of an additional basis function, such as the Difference of Gaussians, becomes imperative.

**Difference of Gaussians (DoG).** The spatial domain expression and Fourier transform of a DoG with a mean of $\mu$, a variance of $\sigma^2$ and a step size $\nu$ of the variance is shown below:

$$e^{-\frac{(x-\mu)^2}{2\sigma^2}} - e^{-\frac{(x-\mu)^2}{2\left(\frac{\sigma^2}{\nu}\right)}} \leftrightarrow e^{-i\omega\mu}\left(e^{-\frac{\omega^2\sigma^2}{2}} - e^{-\frac{\omega^2\sigma^2}{2\nu}}\right). \tag{2}$$

As depicted in (2), DoG serves as a band-pass filter, with the center frequency being contingent upon $\nu$, as illustrated in Fig. 2 for simulation. Leveraging DoG as the basis function facilitates the effective reconstruction of scene details at the desired frequency by adjusting $\nu$.

The Gaussian function excels in reconstructing low-frequency information, while the GEF demonstrates proficiency in capturing smooth regions with minimal loss, even with a reduced number of functions. On the other hand, the DoG proves effective in re-constructing scene details. Leveraging the distinctive strengths of these three basis functions, we introduce Tangram-Splatting. By harnessing the low-pass property of Gaussians for large-scale scene reconstruction, coupled with the shape adaptability of GEF and the band-pass nature of DoG for fine-tuning, we achieve a remarkable reduction in memory usage by 62.4% and a 44.2% increase in computational speed compared to 3DGS [14]. Notably, this improvement is achieved without resorting to a plethora of small Gaussian spheres, resulting in a more compact scene reconstruction.

## 3.2 Signal Estimation Modeling

To enhance the computational efficiency of 3DGS [14], we adopt a novel approach in Tangram-Splatting, eschewing the construction of three distinct models for scene reconstruction. Instead, we frame the problem of scene reconstruction with different basis functions as a signal estimation task.

Given that fast rasterization in 3DGS [14] is tailored for Gaussian functions, we conceptualize the field space $\sum_{j=1}^{N} \omega_j e^{-\frac{1}{2}x^T \Sigma^{-1} x}$, formed by Gaussian spheres, as the observation space $\mathbf{Z}$. Although we receive a sample signal of $N$ Gaussian spheres from $\mathbf{Z}$, it often contains noise, such as neural network performance fluctuations or inherent dataset noise, compromising the reconstruction quality. To enhance the accuracy of estimating characteristic parameters from noise-affected samples within the observation space $\mathbf{Z}$, we introduce a parameter space $\theta$, where $\theta_0 = \beta$ and $\theta_1 = \nu$. Here, $\beta$ dynamically adjusts the shape of the GEF, and $\nu$ adaptively tunes the center frequency of the band-pass filter.

Utilizing maximum likelihood estimation, we refine $\hat{\beta}$ and $\hat{\nu}$ from the noise-contaminated samples of $\mathbf{Z}$. Subsequently, we adjust the Gaussian spheres within $\mathbf{Z}$ to compose the estimation space $\mathbf{Y}$, comprising three distinct basis functions: Gaussian, GEF, and DoG. This process, illustrated in Fig. 1, enables more accurate estimation and modification of the observation space, leading to enhanced scene reconstruction in low memory consumption.

## 3.3 Maximum Likelihood Estimation

The maximum likelihood estimate of GEF is discussed in work [4], and this estimate is approximated in work [10] to a smoother function:

$$\phi(\beta) = \frac{2}{1 + e^{-\beta}}, \tag{3}$$

where $\beta \in [0, +\infty)$, (3) is a monotone increasing function. (3) gives an estimate of the covariance in the estimation space, *i.e.*, $\hat{\sigma} = \phi(\beta)\sigma$.

For the maximum likelihood estimation of DoG, we analyzed as follows.

**Theorem 3.1. (Maximum Likelihood Estimation of DoG)** The maximum likelihood estimate of the DoG function with respect

to the covariance:

$$\hat{\sigma} = \sqrt{\frac{\nu - 1}{2\ln(2\nu)}} \sqrt{\sum_{i=1}^{N} x_i^2}. \tag{4}$$

**Proof.** The likelihood function of DoG is as follows, and cross terms in the formula are removed for easy calculation:

$$L(x; \sigma) = e^{-\frac{\left(\sum_{i=1}^{N}(x_i - \mu)^2\right)}{2\sigma^2}} - e^{-\frac{\left(\sum_{i=1}^{N}(x_i - \mu)^2\right)}{2\left(\frac{\sigma^2}{\nu}\right)}}. \tag{5}$$

Then set $\mu = 0$ and take the logarithm of both sides:

$$\ln L(x; \sigma) = \ln \left[ e^{-\frac{\left(\sum_{i=1}^{N} x_i^2\right)}{2\sigma^2}} - e^{-\frac{\left(\sum_{i=1}^{N} x_i^2\right)}{2\left(\frac{\sigma^2}{\nu}\right)}} \right]. \tag{6}$$

Take the derivative of both sides with respect to $\sigma$:

$$\frac{\partial \ln L(x; \sigma)}{\partial \sigma} = \frac{e^{-\frac{\sum_{i=1}^{N} x_i^2}{2\sigma^2}} \frac{\sum_{i=1}^{N} x_i^2}{\sigma^3} - e^{-\frac{\sum_{i=1}^{N} x_i^2}{2\left(\frac{\sigma^2}{\nu}\right)}} \frac{2\nu \sum_{i=1}^{N} x_i^2}{\sigma^3}}{e^{-\frac{\sum_{i=1}^{N} x_i^2}{2\sigma^2}} - e^{-\frac{\sum_{i=1}^{N} x_i^2}{2\left(\frac{\sigma^2}{\nu}\right)}}}. \tag{7}$$

Let $\frac{\partial \ln L(x;\sigma)}{\partial \sigma} = 0$, we have:

$$\hat{\sigma} = \sqrt{\frac{\nu - 1}{2\ln(2\nu)}} \sqrt{\sum_{i=1}^{N} x_i^2}. \tag{8}$$

Since $\sqrt{\sum_{i=1}^{N} x_i^2}$ is a statistic of $\sigma$, we have the estimation formula:

$$\phi(\nu) = \sqrt{\frac{\nu - 1}{2\ln(2\nu)}}. \tag{9}$$

Since the domain of $\phi(\nu)$ does not contain 0 and is not differentiable at $\frac{1}{2}$, in order to make the formula smooth, we let $\nu \in [2, +\infty)$.

Through $\phi(\beta)$ and $\phi(\nu)$, we can use these two formulas to modify the covariance of the Gaussian function in the observation space $\mathbf{Z}$, and obtain the covariance under GEF and DoG respectively, so as to construct the estimation space $\mathbf{Y}$ of the span of the three basis functions. Using an approach similar to work [10], we set $\hat{S}(\theta) = \phi(\theta)S$, which has the same benefit as directly modifying the covariance matrix $\Sigma$ of 3DGS [14]. Because the covariance matrix $\Sigma$ is a real symmetric semidefinite matrix. $\Sigma = RSS^T R^T$, where $R$ is the identity orthogonal matrix, $S$ is the diagonal matrix, and the elements in $S$ are the root of the eigenvalues. The scaling matrix $S$ controls the degree of anisotropy of the covariance matrix.

## 4 METHODS

### 4.1 Preliminary

The geometry of 3DGS [14], begins with the mathematical expression of a 3D Gaussian function:

$$G(x) = \frac{1}{(2\pi)^{\frac{d}{2}} |\Sigma|^{\frac{1}{2}}} \exp\left(-\frac{1}{2}(x - \mu)^T \Sigma^{-1} (x - \mu)\right). \tag{10}$$

It then describes how 2D projection is achieved through a viewing transformation $(W)$ and the Jacobian $(J)$ of an affine projective transformation:

$$\Sigma' = JW\Sigma W^T J^T. \tag{11}$$

Additionally, it explains the positive semi-definite covariance of the system, where the covariance matrix ($\Sigma$) is decomposed into the product of scale ($S$) and rotation ($R$) matrices, denoting positive semi-definite covariance:

$$\Sigma = RSS^\mathrm{T}R^\mathrm{T}. \tag{12}$$

3DGS [14] leverages the 3-dimensional Gaussian functions, rasterization methods and adaptive control (including cloning, splitting and pruning methods) to approximate a 3D scene.

## 4.2 Diversification based on Shape Priors

The *densify and clone* operation in 3DGS [14] relies on thresholds for position gradient and maximum value of the scaling matrix $S$. 3DGS [14] suggests that large position gradients indicate poor geometric reconstruction, while small maximum values of $S$ trigger cloning of small Gaussian spheres, potentially leading to excessive memory usage and reduced computational efficiency. This highlights the need to enhance the *densify and clone* operation to address under-reconstruction issues.

In order to make the representation more compact, we allocate the Gaussian form based on the shape priors. For the under-reconstruction region, we avoid cloning a large number of small Gaussian spheres by adding two basis functions, GEF and DoG, on the basis of Gaussian function. The introduction of these two basis functions with different properties can make our Tangram-Splatting algorithm more compact in reconstruction like a Tangram puzzle, thus compressing memory. We define the DoG formula for position $x$ and the symmetric positive definite covariance matrix $\Sigma$ in three-dimensional space as follows:

$$D(x) = e^{-\frac{1}{2}x^\mathrm{T}(RSS^\mathrm{T}R^\mathrm{T})^{-1}x} - e^{-\frac{1}{2}x^\mathrm{T}\left(R\left(\frac{S}{\nu}\right)\left(\frac{S}{\nu}\right)^\mathrm{T}R^\mathrm{T}\right)^{-1}x}, \tag{13}$$

where $R$ is the rotation matrix of $\Sigma$ and $S$ is the scaling matrix of $\Sigma$. $\nu$ is the step length between covariance of two Gaussians, always larger than 2, and is updated by the neural network, while $\nu$ also controls the anisotropic properties and center frequency of the DoG. The GEF formula for position $x$ and covariance matrix $\Sigma$ in three-dimensional space is defined in [10]:

$$L(x) = e^{-\frac{1}{2}(x)^\mathrm{T}\Sigma^{-1}(x)^{\frac{\beta}{2}}}, \tag{14}$$

where $\beta$ is the shape parameter of GEF and updated by network. Thus, Eq. (10), (13), and (14) constitute the three basis function representations of Tangram-Splatting.

## 4.3 Defination of the Adaptive Matrix

Inspired by the signal estimation theory in Sec. 3.2 and [10], we do not need to explicitly construct three models for Eq. (10), (13), and (14). We only estimate and modify the parameters of the covariance matrix of the observation space $\mathbf{Z}$ through the maximum likelihood theorem in the estimation space $\mathbf{Y}$. This method not only saves the training time, but also looks like the three basis functions spanning the estimation space.

Since the essence of the modified covariance matrix $\sigma$ is to correct the anisotropy property of the Gaussian function, and the anisotropy property is reflected by the eigenvalues of the matrix, we only need to correct the scaling matrix $S$ to achieve this purpose

(because $S$ is a diagonal matrix formed by the square root of the eigenvalues). The correction method is defined by the following two formulas:

$$\hat{S}(\beta) = \phi(\beta)S, \tag{15}$$

and

$$\hat{S}(\nu) = \phi(\nu)S. \tag{16}$$

Considering that it is necessary to increase the number of points appropriately to achieve a better reconstruction effect in complex scenes, we perform *densify and clone* operations in the first $n$ iterations. This allows the algorithm to use the original Gaussian spheres to reconstruct the scene over a wide coarse-grained range. Then, in order to avoid the scene being filled with a lot of small Gaussian spheres, instead of *densify and clone* operations, we replace the Gaussian spheres that satisfy the clone condition with GEF or DoG which are optimized by the network. The choice of GEF or DoG depends on the degree of anisotropy of the Gaussian spheres satisfying the cloning condition. When the largest eigenvalue of the matrix $S$ is larger than $k$ times of the second largest eigenvalue (we set $k = 5$), we consider its anisotropy property to be significant, so the DoG with adaptive center frequency is used to replace the Gaussian spheres. If not, GEF is used instead.

## 5 EXPERIMENTS

### 5.1 Datasets and Metrics

To assess the efficacy of our approach in memory compression and training speed enhancement, we utilized the eigenvalues of the Gaussian spheres as shape priors to construct three types of basis functions and conducted experiments utilizing a diverse array of real-world scene datasets. Specifically, we focused on two scenes sourced from the Tanks&Temples dataset [16], nine scenes from the Mip-NeRF360 dataset [2], and two scenes provided in Deep Blending [11]. These datasets were deliberately selected to align with those utilized in the evaluation of 3DGS [14], ensuring comparability and facilitating a comprehensive assessment of our method's performance.

We compared the average training time and memory consumption required to store optimized parameters after 40,000 training iterations. It is important to note that the stored parameters of point clouds remain consistent with those used in 3DGS [14]. Furthermore, we employed standard metrics such as PSNR, L-PIPS, and SSIM to quantitatively evaluate the reconstruction quality of test views. To maintain consistency and avoid confusion regarding the current state-of-the-art (SOTA) methods, we extracted numerical results from previous works [2, 10, 14] and included them in Table 1.

### 5.2 Implementations

We utilized an A100 GPU for the majority of our tests. The learning rate for the step parameter of the covariance of the DoG was set to 0.00025, with a step reset interval of 1000 iterations. The learning rate for the shape parameter of the GEF, as well as the shape reset interval and the density gradient threshold, were set according to the specifications in GES [10]. Other hyperparameters and design choices were consistent with those of 3DGS [14]. Additionally, to

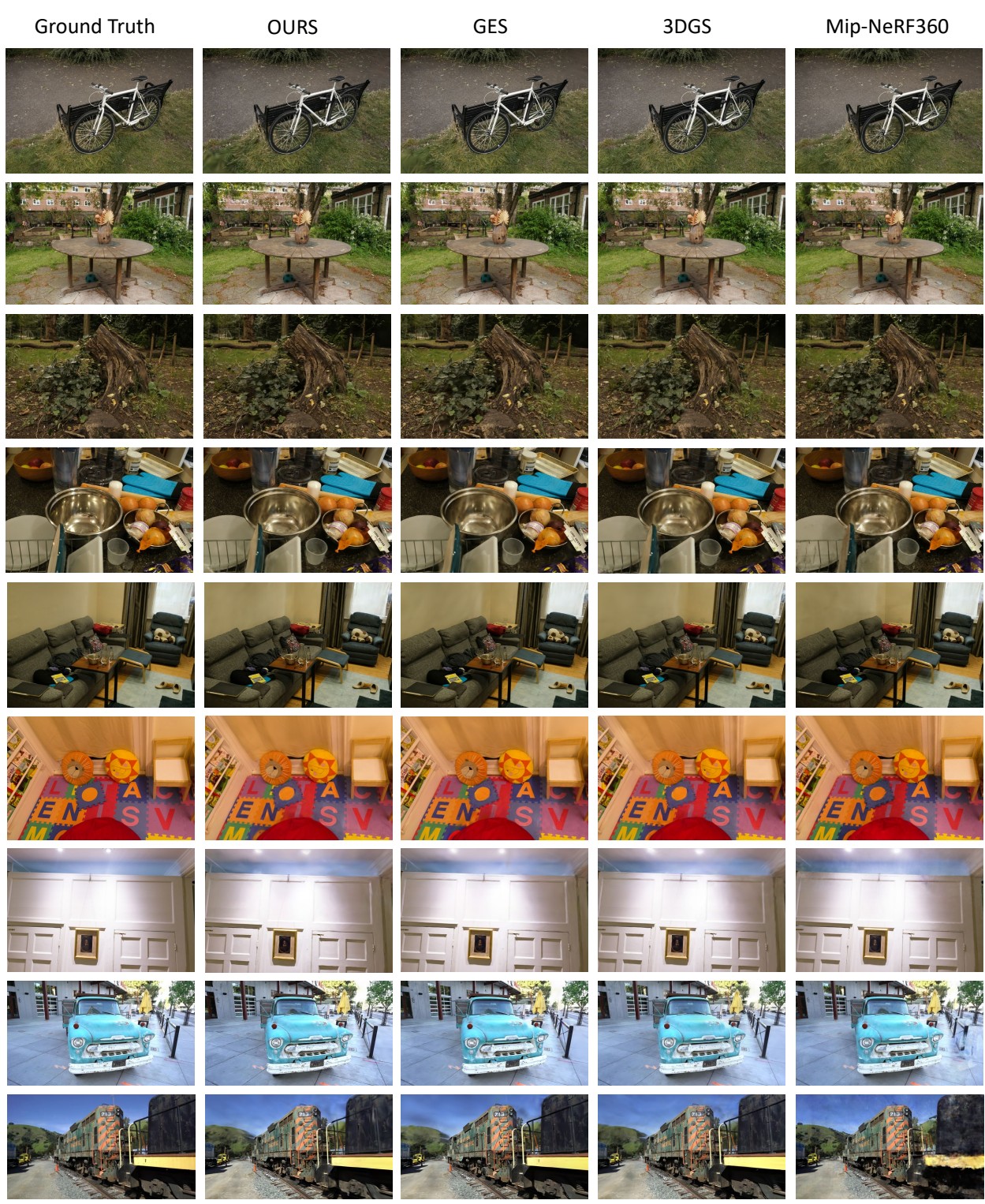

**Figure 3: Qualitative Results. We compare our method to GES[10], 3DGS [14] and Mip-NeRF360[2]. From the top to bottom are the scenes BICYCLE, GARDEN, STUMP, COUNTER, ROOM from the Mip-NeRF360 dataset [2], PLAYROOM and DRJOHNSON from the Deep Blending [11], TRUCK and TRAIN from the Tanks&Temples dataset [16]. The reconstruction performances of Tangram-Splatting are comparable to GES [10] and 3DGS [14].**

| | Mip-NeRF360 Dataset | | | | | Tanks&Temples | | | | | Deep Blending | | | | |
|---|---|---|---|---|---|---|---|---|---|---|---|---|---|---|---|
| | SSIM↑ | PSNR↑ | LPIPS↓ | Train↓ | Mem↓ | SSIM↑ | PSNR↑ | LPIPS↓ | Train↓ | Mem↓ | SSIM↑ | PSNR↑ | LPIPS↓ | Train↓ | Mem↓ |
| Plenoxels | 0.626 | 23.08 | 0.463 | 26m | 2.1GB | 0.719 | 21.08 | 0.379 | 25m | 2.3GB | 0.795 | 23.06 | 0.510 | 28m | 2.7GB |
| INGP | 0.699 | 25.59 | 0.331 | 7.5m | 48MB | 0.745 | 21.92 | 0.305 | 7m | 48MB | 0.817 | 24.96 | 0.390 | 8m | 48MB |
| Mip-NeRF360 | 0.792 | 27.69 | 0.237 | 48h | 8.6MB | 0.759 | 22.22 | 0.257 | 48h | 8.6MB | 0.901 | 29.40 | 0.245 | 48h | 8.6MB |
| 3D Gaussians-7K | 0.770 | 25.60 | 0.279 | 6.5m | 523MB | 0.767 | 21.20 | 0.280 | 7m | 270MB | 0.875 | 27.78 | 0.317 | 4.5m | 386MB |
| 3D Gaussians-30K | 0.815 | 27.21 | 0.214 | 42m | 734MB | 0.841 | 23.14 | 0.183 | 26m | 411MB | 0.903 | 29.41 | 0.243 | 36m | 676MB |
| GES-40K | 0.794 | 26.91 | 0.250 | 32m | 377MB | 0.836 | 23.35 | 0.198 | 21m | 222MB | 0.901 | 29.68 | 0.252 | 30m | 399MB |
| Tangram-Splatting (ours) | 0.793 | 26.95 | 0.225 | 20m | 298MB | 0.819 | 23.22 | 0.218 | 15m | 150MB | 0.902 | 29.59 | 0.253 | 23m | 236MB |

**Table 1: Quantitative Results. This table provides a thorough comparison between our approach and established methods across diverse datasets. By incorporating metrics such as SSIM, PSNR, and LPIPS, alongside training duration and memory usage, it offers a comprehensive view of performance effectiveness. It is important to acknowledge that the training times for different methods may have been calculated on different GPUs, which could affect comparability, but the results remain valid. Please note that implicit representations, such as INGP [23] and Mip-NeRF360 [2], have limited memory as they rely on slower neural networks for decoding. Notably, superior performance is highlighted in red.**

accommodate differences in texture details among images from various datasets, we allowed *densify and clone* operations for different scenes with fewer iterations, as elaborated in the *Appendix*.

## 6 RESULTS

### 6.1 Quantitative Results

We compared our method with 3DGS [14], GES [10], Mip-NeRF360 [2], InstantNGP [23] and Plenoxels [6] on the Tanks&Temples [16] dataset, the Mip-NeRF360 dataset [2] and two scenes provided in Deep Blending [11]. It is important to note that our Tangram-Splatting approach primarily focuses on leveraging basis functions with distinct properties to achieve a more compact representation. As such, we only adopted the GEF basis functions from the GES algorithm [10], and did not incorporate other innovative enhancements introduced in GES [10] into our algorithm. The quantitative results are presented in Table 1.

**Compactness.** From the perspective of compact representation, we compared our Tangram-Splatting algorithm, utilizing three different types of anisotropic basis functions, with 3DGS [14] using solely anisotropic Gaussian spheres, and GES [10] employing only GEF as the basis function. As illustrated in Table 1, for the Tanks&Temples dataset, our method incurs a memory cost after 40,000 iterations that is 0.365 times that of the memory produced by 3DGS [14], and 0.676 times that of GES [10]. Our PSNR is increased by 0.08 compared to 3DGS [14] while being decreased by 0.13 as compared to GES [10], the difference is marginal.

Given the finer texture of scenes in the Mip-NeRF360 dataset [2], we allowed certain iterations of density and clone operations for certain scenes based on our experiences (*e.g.*, BICYCLE, STUMP, and GARDEN). Even with this adjustment, the memory cost after 40,000 iterations remains significantly lower, at 0.406 times and 0.790 times that of 3DGS [14] and GES [10], respectively, albeit with a slight decrease in PSNR by about 0.26 compared to 3DGS [14] and an increase of 0.04 compared to GES [10].

Regarding the scenes from Deep Blending [11], the PSNR is increased by 0.18 compared to 3DGS [14] and only experiences an average decrease of 0.09 compared to GES [10], while the memory cost after 40,000 iterations is merely 0.349 times and 0.591 times that of 3DGS [14] and GES [10], respectively.

In Table 1, it is evident that our algorithm significantly reduces memory consumption while maintaining reconstruction quality. Comparatively, our compression ratio indicates a remarkable reduction of 62.4% in memory usage compared to 3DGS [14], with GES [10] achieving a slightly lower reduction of 43.2%. In terms of PSNR, our algorithm's performance falls between that of 3DGS [14] and GES [10], with marginal deviations being observed.

**Computational efficiency.** By reducing the number of point clouds, minimizing the computation of numerous small Gaussian spheres, and streamlining *densify and clone* operations, our method significantly decreases training time across the three aforementioned datasets. Additionally, in the replacement operations introduced in Tangram-Splatting between different basis functions, we optimize the process by replacing the computation of eigenvalues of the covariance matrix $\Sigma$ with direct sorting of the scaling matrix $S$, leveraging the mathematical properties of the covariance matrix. This enhancement further contributes to the reduction in training time. As indicated in Table 1, the training time for Tangram-Splatting is decreased by an average of 22 minutes, 11 minutes, and 13 minutes, respectively, compared to 3DGS [14].

### 6.2 Qualitative Results

While compressing memory, we also need to consider the reconstruction effect of Tangram-Splatting. The qualitative results on the Tanks&Temples dataset [16] , the Mip-NeRF360 dataset [2] and the two scenes provided in Deep Blending [11] are shown in Fig. 3. The results show that the overall effect of Tangram-Splatting is comparable to the reconstruction effect of previous work, which is hard to tell the differences of rendered images between Tangram-Splatting and previous work. Only in some details (such as reflective Windows in the GARDEN scene from the Mip-NeRF360 dataset [2], shadows in the sky in the TRAIN scene from the Tanks&Temples dataset [16]) the performances are lower than the Ground Truth.

### 6.3 Analysis of point clouds structure

We show the point clouds of the scene FLOWERS, COUNTER, KITCHEN and ROOM from the Mip-NeRF360 dataset [2] from top to bottom, as shown in Fig. 4. We showcase the comparisons of point clouds produced by our Tangram-Splatting and 3DGS [14]. We observe that both Tangram-Splatting and 3DGS [14] effectively

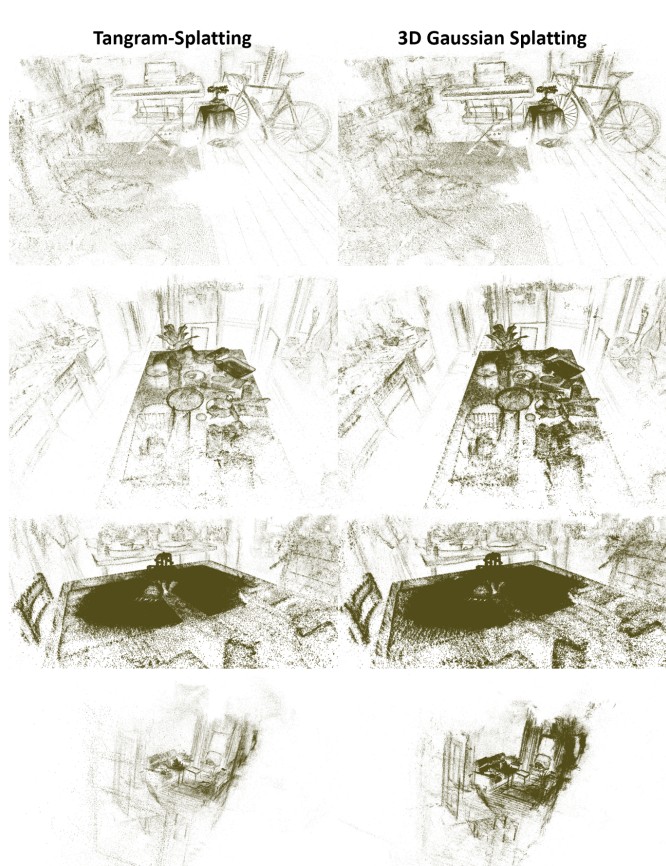

**Tangram-Splatting**    **3D Gaussian Splatting**

**Figure 4: Point clouds Comparison. In order to check the difference between Gaussian distribution, we render the point clouds result to make a comparison. The left hand side are the results of Tangram-Splatting while the right hand side are the results of 3DGS [14]. We can infer from the images that our method can reduce the number of Gaussians needed in reconstruction while maintaining the PSNR performance.**

capture the outline contour of the 3D scene. However, Tangram-Splatting achieves this with a significantly smaller number of Gaussian counts, achieved by reducing redundant Gaussians that are unnecessary for 3D reconstruction. We observe that our point clouds are more sparsely distributed in specific areas: the white wall behind the bicycle in the FLOWERS scene and the white wall behind the sofa in the ROOM scene. This suggests that Tangram-Splatting tends to reconstruct flat areas with fewer functions but larger volumes.

## 6.4 Ablations

We re-run the GES [10] algorithm with the same parameter settings as Tangram-Splatting to highlight the robustness of our algorithm by comparing the reconstruction performance with approximately the same memory. To ensure memory consistency and accurately

|  | Ablation Study | | | |
|---|---|---|---|---|
|  | SSIM$^\uparrow$ | PSNR$^\uparrow$ | LPIPS$^\downarrow$ | Mem$^\downarrow$ |
| GES-40K | 0.903 | 29.578 | 0.253 | 305MB |
| Ours | 0.902 | 29.591 | 0.253 | 236MB |

**Table 2: Ablation Results. We do the ablation study on the Deep Blending dataset [11]. We try to make the Gaussian count of GES [10] to our algorithm to see whether our representation is better than GES's [10] (whose paper has already shown it is better than 3DGS's [14]). We can indicate in the table that our algorithm is a better representative method, when our performance is slightly better and still reduces a lot of memory cost.**

Ground Truth    OURS    GES

**Figure 5: Comparison of ablation experiments. We manage memory consumption consistently and showcase the DR-JOHNSON scene to demonstrate how the introduction of basis functions with diverse properties enriches detail reconstruction. Our Tangram-Splatting excels in reconstructing the slit of the table, while the effect of GES [10] appears blurry and includes white artifacts.**

assess the reconstruction performance of both algorithms, we maintain the density iterations of GES [10] in line with those of Tangram-Splatting, while leaving the remaining parameters of GES [10] unchanged. We evaluate the entire scenes from the Deep Blending dataset [11]. The quantitative results presented in Table 2 showcase that, despite our lower memory consumption, our performance consistently excels. Furthermore, the qualitative results, illustrated in Fig. 5, highlight the proficiency of our method in reconstructing intricate details, such as straight lines. We observe that the lines in the magnified region of GES [10] appear more blurred compared to ours, and there is a pronounced white artifact present in the lower left corner, as shown in Fig. 5. This enhancement can be attributed to the inherent anisotropy of GEF and DoG.

## 7 CONCLUSION AND FUTURE WORK

In this paper, we put forward a new method for a more compact representation for 3D reconstruction. We leverage signals and systems perspective to make the base function more compact than vanilla Gaussian function. However, our method somewhat revisits the problem from the last century of designing base functions in the field of signal processing. In the future work, we may borrow the idea of implicit representation from Neural Radiance Field [21] to use networks to represent a base function (while maintaining the same speed) instead of manually designing a new compact base function.

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
