# OpenReview forum: "Tangram-Splatting: Optimizing 3D Gaussian Splatting Through Tangram-inspired Shape Priors"
_acmmm.org/ACMMM/2024/Conference — MM2024 Poster_

### Official Review · Reviewer_mN55 · 2024-05-20

**Rating:** 5
**Confidence:** 4

**Summary:**

The method proposes a tangram-like Gaussian Splatting strategy to represent the target scene more efficiently by using three different types of Gaussians simultaneously. This strategy can reduce the training time and storage overhead without losing the rendering quality.

**Strengths:**

1. By using Gaussians of different types, redundancy can be reduced when representing a scene, making the representation more compact and efficient.
2. For the training time, since it optimizes the number of Gaussians based on the shape prior，
the method does not increase the training time but rather reduces it, although it introduces other computations.

**Limitations:**

This hybrid representation is very interesting and could be a future trend, but there are still some problems in terms of this paper:
1. This paper does similar work as the Gaussian lightening methods with the same purpose of removing redundant Gaussians or their parameters, so I think it would be better to compare the lightening methods in the experiments to emphasize the advantages of the method to the lightening methods in terms of the training time or the rendering quality.

2. It can be understood that the GEF is used to express the areas with smooth color change, but as long as it changes its β, it can also be turned into a Gaussian distribution with a sharp peak. So, wouldn't it already be able to express the two different cases (smooth color change and dramatic color change) by itself? So why is the original Gaussian still needed, which is missing some analysis?
In addition, the paper claims that GEF loses its Gaussian property in the frequency domain, which could lead to non-convergence, and therefore, GEF is problematic. So why does the paper still use GEF and doesn't modify it, which lacks an explanation?

3. The paper didn't show the effect of DoG in ablation. As the authors said, DoG is more likely to express scene details. However, a Gaussian can only carry one color, so it still can't express too complex textures even if it has a more complex spatial domain distribution. Maybe it can express repetitive textures better. Thus, I doubt whether this DoG is useful or not.
In addition, in the ablation study, the authors should give some visual results to show more intuitively what scenarios different types of Gaussians are more inclined to express and whether they are consistent with what the authors analyzed in section 3.

4. Also, the paper doesn't prove that the types of Gaussians chosen are reasonable. While the idea of using different types of Gaussians simultaneously is correct, I don't think the artificial selection of the types of Gaussians is reasonable.

5. Fig.3 shows little difference. It would be better to modify this figure to reflect the advantages of the proposed method.

**Suitability:**

3

---

### Official Review · Reviewer_HUmK · 2024-05-20

**Rating:** 4
**Confidence:** 2

**Summary:**

This paper introduces Tangram-Splatting, an approach for 3D scene reconstruction using different basis functions. By leveraging the strengths of Gaussian, Generalized Exponential, and Difference of Gaussians functions, Tangram-Splatting achieves a more compact representation of the scene with reduced memory usage and increased computational speed compared to existing methods. The article also provides theoretical analyses of the properties of these basis functions and highlights the advantages of using diverse functions for scene reconstruction.

**Strengths:**

The advantages of this paper can be summarized as follows:

1. Innovation: The paper proposes a novel approach by using different types of base functions to optimize the 3D Gaussian Splatting algorithm. This method allows for a more compact representation of 3D scenes compared to traditional Gaussian function representations.

2. Theoretical analysis: The paper provides theoretical analyses of different types of base functions, revealing their properties in the time and frequency domains. This analysis helps to better understand the selection and usage of base functions and their role in the reconstruction process.

**Limitations:**

There are some comments for this paper:
1. The authors claim that the proposed method performs well in detail reconstruction. It seems difficult to distinguish between "GES" and "ours" in Figure 3. It would be beneficial to enhance the visualization by zooming in on specific details in Figure 3.
2. The authors do not mention the loss function. For GES, it gives the design of the loss function and conducts ablation studies for it.
3. I would like to know how many training iterations were used for the other comparison experiments, as I understand that training iterations can have an impact on the experimental results. Additionally, I'm interested in understanding the experimental settings of the comparative methods. Could you please provide me with this information?
4. The author claims that this method can reduce memory usage and, compared to GES-4K, there might be a slight performance drop. Would it be possible to compare the memory usage at the same performance level?

suggestion:
1. The author should provide some explanations about the models "3D Gaussians-7K" and "3D Gaussians-30K" for better understanding. Different numbers represent different iterations number？
2.  The authors should provide explanations for all the colors used in Table 1.

**Suitability:**

2

---

### Official Review · Reviewer_5fBs · 2024-05-24

**Rating:** 4
**Confidence:** 4

**Summary:**

This paper proposes a hybrid primitive representation to improve the compactness of 3D Gaussian splatting. Specifically, in 3D Gaussians, and generalized exponential functions (GEF) are leveraged to represent low-frequency signals of the scene and differences of Gaussians (DoG) are used as band-pass filters to effectively represent objects with sharp shapes. Experimental results demonstrate that the proposed hybrid primitive representation can effectively reduce the number of primitives required for scene modeling.

**Strengths:**

1. The proposed hybrid primitive representation is technically reasonable, as this representation can leverage the advantage of three types of primitives to handle the distinct properties of different objects.
2. The theoretical analysis in Sec.3 is comprehensive and detailed.
3. Experimental results show that the proposed hybrid representation can significantly reduce the model size and training time, which demonstrates the effectiveness of the proposed method.

**Limitations:**

1. The smoother function of GEF shown in Eq. 3 is different from the form in work [10], wherein the function in work [10] has the hyper-parameter $/rho$
2. It would be better to add more discussion and details about the derivation of Eq. 4. For example, authors should discuss the rationale for ignoring cross terms in the likelihood function of DoG.
3. Authors should explain the definition of the field space $\sum_{j=1}^{N}\omega_j e^{-\frac{1}{2}\boldsymbol{x}^T\Sigma^{-1}\boldsymbol{x}$. For example, the definition of $\omega_j$ should be clarified.
4. It would be better to identify which primitives belong to DoG and which primitives belong to other types of primitives in Fig. 4.

**Suitability:**

3

---

### Official Review · Reviewer_Ptoa · 2024-05-25

**Rating:** 4
**Confidence:** 4

**Summary:**

The paper introduces a novel approach to 3D reconstruction, incorporating shape priors inspired by the Tangram puzzle, an ancient Chinese puzzle, to optimize the 3D Gaussian Splatting method to improve memory efficiency and reduce computational overhead in 3D scene reconstruction. The approach integrates different Gaussian functions to refine scene fitting, achieving notable reductions in memory usage and training time. The paper presents a comprehensive evaluation of the proposed method, Tangram-Splatting, against existing techniques using various datasets and metrics.

**Strengths:**

1. The paper presents an innovative approach to 3D reconstruction by introducing shape priors inspired by tangram, which is a unique perspective in the field.
2. The paper provides a thorough empirical evaluation, demonstrating significant memory reduction and improved training time, which are crucial for practical applications.

**Limitations:**

Insufficient evaluation:
1. The rationale for incorporating both the Gaussian function and GEF, along with DoG, into the proposed Tangram-Splatting approach is not entirely clear. Given that the Gaussian function is a specific instance of GEF,  it raises the question of why both are utilized concurrently. Ideally, GEF should encompass the functional capabilities of the Gaussian function by adjusting $\beta$  accordingly.  The paper could benefit from a deeper exploration or justification for the simultaneous use of these specific functions, particularly focusing on whether the Gaussian function introduces any unique benefits that a variable $\beta$ in GEF does not already provide.

2. Clarification of Step Size:
    1). The justification provided for restricting the parameter $\nu$ to the range $[2, +\infty)$ is somewhat ambiguous. The paper states that this range is chosen because the function $\phi(\nu)$ is not differentiable at $\nu = \frac{1}{2}$ and does not contain 0. However, the connection between these mathematical properties and the practical implications for choosing $\nu \geq 2$ needs to be more clearly articulated.
    2). Additionally, the impact of varying step size $\nu$ on the reconstruction quality and efficiency is a critical aspect that seems underexplored.

3. Proportion of Each Basis Function
    The manuscript could benefit from including detailed information about the proportion in which each basis function is utilized. Detailed statistical or visual representations of their usage across various datasets or scenes would provide clearer insights into the method’s operational dynamics.

4. Necessity of DoG in Ablation Study:
 An ablation study that methodically examines the reconstruction performance with and without the DoG component would offer a clearer perspective on its efficacy. The baseline method GES does not include Gaussian function and DoG, and thus the comparison between GES can not directly demonstrate the efficacy of DoG.

Lack of novelty:
5. Given the weaknesses highlighted in points 1 and 4, the method may not sufficiently demonstrate new contributions over existing work. (Compared to the baseline method GES)

Minor:
1. Figure 3 would benefit from the addition of visual quality metrics directly on each rendered image. This would provide a clearer, immediate understanding of the performance differences between the presented methods.
2. The captions for figures should be more concise. Descriptive details of results should be relocated to the main text to allow captions to focus solely on describing the visual content.

**Suitability:**

3

---

### Meta-Review · Area_Chair_AcNs · 2024-06-27

**Recommendation:** Accept (Poster)
**Confidence:** 4

**Metareview:**

The reviewers unanimously voted to accept this paper. Therefore, we are happy to recommend the acceptance of this paper.